# Measuring Athletes’ Perception of the Sport Nutrition Information Environment: The Adaptation and Validation of the Diet Information Overload Scale among Elite Athletes

**DOI:** 10.3390/nu13082781

**Published:** 2021-08-13

**Authors:** Anna Kiss, Sándor Soós, Orsolya Tompa, Ágoston Temesi, Zoltán Lakner

**Affiliations:** 1Faculty of Education and Psychology, Eötvös Loránd University, 1075 Budapest, Hungary; soossand@gmail.com; 2Department of Science Policy and Scientometrics, Library and Information Centre of the Hungarian Academy of Sciences, 1051 Budapest, Hungary; 3Department of Food Chain Management, Institute of Agribusiness, Faculty of Economics and Social Sciences, Hungarian University of Agriculture and Life Sciences, 2100 Gödöllő, Hungary; orsolya.tompa@olympian.org (O.T.); temesi.agoston@uni-mate.hu (Á.T.); lakner.zoltan@uni-mate.hu (Z.L.)

**Keywords:** information overload, factor analysis, sport nutrition environment, preventive behavior

## Abstract

The exponentially growing quantity of nutrition information creates a new situation and challenge for every stakeholder, from athletes, coaches and nutritionists to policymakers. To measure the perception of the information environment related to healthy eating, the diet information overload scale was developed. The scale consists of eight items, measuring the perceived importance of pieces of information overload on Likert-type scales. The objective of the study was to test the applicability and validity of the diet information overload scale among athletes. A cross-sectional validation study was conducted with elite athletes (*n* = 177). To validate each item of the scale, we applied Cronbach’s alpha test, and the inner consistency of the scale was analyzed with linear correlation coefficients of the different variables. To evaluate the relationship between question groups, we applied factor analysis. The different fit indices showed a good fit to the model; the Root Mean Square Error of Approximation (RMSEA) value was 0.09 and the Tucker–Lewis index (TLI) value was 0.84. The indicators of reliability (α based upon the covariances = 0.81) produced suitable results; thus, the sport nutrition information overload scale showed high reliability and applicability. Based on the sport nutrition information overload scale, further analysis could be carried out on how to optimize the content of key pieces of sport nutrition-related information.

## 1. Introduction

Western philosophy has been based on the paradigm that “information is good in virtue of being information” [1]. It has been proven that an informed citizen makes better choices [2]. The practical importance of academic and rigorous analysis of the information process is constantly growing because (1) the abundance of information and the relative ease of its accessibility radically increase and change societies [3] and (2) there is increasing noise in communication channels, due to manipulations [4].

There is increasing attention on information overload. By analyzing the Web of Science database, as one of the most comprehensive, high-quality, leading sources of academic publications [5], we conducted the analysis based on the following search term:

(TS = (“information overload*” AND (“cons*” OR “decision*” OR “psy*”))) AND LANGUAGE: (English).

We have found 1426 articles, published in the English language in the time period of 1975–2021, on this topic. These articles were analyzed by the Vos viewer system, based on the clustering of keywords co-occurrence. The cluster analysis of the co-occurrence of keywords yielded nine clusters, which can be interpreted as the main directions of the research. These are summarized and visualized in Figure 1.

There are three overarching topics, around which numerous research problems are clustered: the communication, informatics, and behavior science aspects of the information overload field. Interestingly, the communication-focused cluster embraces health-related aspects too, but nutrition-related communication does not appear in this cluster as an independent node, as opposed to the cancer-related one. Five applied fields of information overload appear in different clusters: consumer research, management science, social media, marketing, and pedagogy-related fields. One cluster has a relatively heterogeneous character, from COVID-19-related problems to meta-analysis of former communications. This cluster of topics embraces the latest directions of research.

Information overload research is a key, rapidly developing, highly diverse topic, one of the key aspects of which is health-related communication and its efficiency. This can be explained by the fact that the information process plays a key role in the context of health. Non-communicable diseases (NCDs) such as obesity, cardiovascular diseases, cancer, and diabetes are responsible for 70% of total deaths worldwide [6]. To decrease the burden of NCDs, communication campaigns about healthy nutrition and diet could be useful as a method for public interventions [7]. Since nutrition and diet are popular topics, new information about food products or dietary advice often appears in the news and on social media platforms. Exposure to an overwhelming amount of information from the environment full of contradictory facts about health and nutrition can lead to information overload. According to the information overload model, due to the excessive amount of content in the information environment, further cognitive resources are needed to process it. Thus, someone overwhelmed with information will intentionally avoid it. Information overload may cause people to be uncertain and prone to create beliefs about nutrition that negatively affect their preventive behavior [8]. Jensen and co-workers have developed and validated the cancer information overload scale (CIO) to understand and relieve information overload [9]. The CIO has been used by several researchers, in which the association between information environment and health behavior was supported. In their work, they have described that exposure to the information environment predicts information overload and negatively correlates with preventive behavior. To understand the negative effect of exposure to the information environment, Ramondt and Ramírez (2019) extended the concept of information overload to the association of nutritional and dietary information environment that could also be described as loaded with false information and contradictory advice [10]. As a result, they have created the validated diet information overload (DIO) scale.

Athletes can be considered as a vulnerable group of consumers because sport demands an ever-increasing allocation of time and energy [11,12]. Athletes are at the center of the flow of nutrition information [13], but it is an open-ended question as to how they can process these pieces of knowledge. Previously, numerous analyses have been carried out on some general characteristics or particular aspects of nutritional knowledge or education of athletes [14,15], but we do not have a holistic, general overview on the absorption of nutrition information by athletes. The extension of the concept of information overload to the information environment related to sport nutrition—which is also loaded with contradictory statements—has not been studied yet.

The objective of this study was to adapt and validate the diet information overload scale to measure sport nutrition information overload among athletes.

## 2. Materials and Methods

### 2.1. Study Population

Data analyzed in this study were obtained from a cross-sectional survey carried out between 2020 and 2021. The target group consisted of elite and recreational athletes; the inclusion criteria were at least 60 min of physical activity 3 times a week and age above 18 years. Originally, we planned to extend our survey to both recreational and elite athletes, but for the sake of a more homogenous group of athletes, we only included elite athletes in our analyses. We considered as elite athletes those who are licensed at a sports club, take part in at least regional-level competitions, and have at least two years’ experience of playing sport. The recruitment of participants was realized by the “snowball method”, based on different social media platforms (first of all, Facebook), and responses were collected by online Google forms. Considering restrictions due to COVID-19, we created an online form of the questionnaire because there was no possibility of filling it out in person. In the introductory part of the questionnaire, we informed the participants of the purpose and scope of the survey, the administration of the questionnaires, and the data. Participants gave their written informed consent following the Declaration of Helsinki and the Science Ethics Code of The Hungarian Academy of Sciences. The ethical approval to conduct the study was granted by the Institutional Ethics Committee of Department Science Policy and Scientometrics, Library and Information Centre of the Hungarian Academy of Sciences (Project identification number: 12733/1, date of approval: 1 February 2021).

The sample size calculation was based on the recommendation of Everitt [16]. According to the “rule of thumb” (*n*:*p*) rule, the ratio between the number of subjects (*n*) and the number of items of the questionnaire (*p*) should be 10. Since the questionnaire is composed of 8 items, it was necessary to enroll at least 80 participants in the study.

### 2.2. Validation Methodology

The questionnaire used in this cross-sectional survey included questions about sociodemographic characteristics and sport activity besides the sport nutrition information overload scale.

We used the usually applied and accepted international methodology for the validation of questionnaires. As the first step, we translated the DIO scale from English to Hungarian with the help of a certified medical translator; then, we translated the Hungarian version back to the English language. For the next step, we asked Hungarian Ph.D. students using English as their working language to interpret the questions of the scale. We selected those terms that were difficult to translate to Hungarian, then the research team re-evaluated and modified the original translations. In the next step, we re-translated the Hungarian version of the questionnaire to English and asked three foreign Ph.D. students, who obtained their MSc/MA qualification in English language programs and now participate in the international Ph.D. program of the Hungarian University of Agriculture and Applied Life Sciences, to interpret the re-translated questions. As a result, we created a trial questionnaire and asked Ph.D. students to fill it out. Thus, we developed a final version that included no details that could lead to misunderstanding.

### 2.3. The Questionnaire

The sport nutrition information overload questionnaire applied in our current work has been developed in an evolving way. The measurement of information overload has considerable traditions: Jensen et al. (2011) created and validated the CIO scale, based on research on cancer-related information overload [9]. Costa et al. (2014) shortened the original scale: in their opinion, 3 elements out of 8 can be discarded because these items do not question the amount of information, so the term of information overload cannot exactly be described by them [17]. Some years later, Ramondt and Raminez (2019) modified the original questionnaire by Jensen to measure information overload on nutrition [10]. Based on these predecessors, we constructed a sport nutrition information overload questionnaire. This is a novel version of the questionnaire because it focuses on sport nutrition overload, and not on nutrition-related information overload in general. Following the adaptation methodology of Ramondt and Ramírez [10], the adaptation involved changing “diet” or “diet prevention” phrases to “sport nutrition” or “eating for sports”. The scale consists of eight items, measuring the perceived importance of different constraints of application of pieces of information on Likert-type scales.

Table 1 shows the items from the scale for cancer patients, the diet information overload scale adapted for healthy adults, and the items adapted for athletes by our research team.

### 2.4. Statistical Analysis

A multistep, one- and multivariable analysis of data was carried out, consisting of a step-by-step process, in line with the latest results of the validation process. In the first phase of investigations, we determined the descriptive characteristics of the responses to different items. Then, we tested the normality of individual responses by the Lilliefors (Kolmogorov–Smirnov) and the Shapiro–Wilk tests. As the first step of validation, we determined the Cronbach α value, as a standard, widely used indicator of item validity. In the next phase, we analyzed the reliability of the scales, if the items are dropped out, one by one. The reliability of the scales has been measured based on Cronbach’s α and Guttman’s λ6. Both reliability indices were calculated based on covariances (“raw” reliabilities) and correlations (standardizes reliabilities). In the last phase of the investigations, a unidimensionality test was fulfilled by confirmatory factor analysis, and based on confirmatory factor analysis, we were able to determine a model.

An alternative possibility of general factor saturation is the omega hierarchical test (ωh) proposed by Roderick P. McDonald [18]. We applied a rather complex method, taking into consideration the specific features of the sample. First, we made a factor analysis of the original dataset, then rotated the factors. We applied the oblique rotation method because this is more suitable to study problems in social sciences than the generally applied varimax algorithm. After all, in the case of an oblique solution, we allow correlation between the factors and do not suppose orthogonality [19]. In the next phase, based on Schmid–Leiman transformation [20], we rotated the factors, and in this way, a general, second-level factor could be determined from the primary factor structure. In this phase, we applied the built-in algorithms of the psych r-package [21]. According to Zinbarg et al. (2006), McDonald’s omega is better to estimate reliability than Cronbach’s α [22]. At the end of the analysis, results were cross-validated by hierarchical cluster analysis. The key steps of the questionnaire’s development, data collection and statistical analysis are summarized in Figure 2.

## 3. Results

### 3.1. Number of Participants and Their Characteristics

Overall, 177 elite athletes participated in the study. The social characteristics of the respondents show that the sample can be considered as a representative sample on the basis of gender. The proportion of persons with the highest qualification is greater compared to the general Hungarian population. The level of sport activity was rather high: the average number of practices per week was 9.3. The athletes involved in the survey participated at sport competitions an average of 17.4 times in a year. They perform 38 types of sport: 21% swimmers, 16.7% canoers and rowers and 14% volleyball players made up the majority of the sample. We inserted a specific question to obtain information on the degree of freedom of participants in the formation of their nutrition. The majority (85%) have indicated that they have a high or very high level of freedom in this matter, on the other hand, 8.1% of respondents indicated that their choices are rather limited. The sociodemographic and sport activity characteristics of the participants are shown in Table 2.

### 3.2. Descriptive Statistics of Responses to Different Items

Most of the respondents accepted the statement that “There is not enough time to do all the things recommended eating for sports”. Seventy five percent of respondents endorsed this factor by at least four on the five-point scale. A similar distribution of opinions characterized the item “No one could actually do all of the sport nutrition recommendations that are given”. The choosing of appropriate recommendation (“There are so many recommendations about sport nutrition, it’s hard to know which ones to follow”) item was evaluated as an important problem for more than half of the respondents, but in this case, 25% evaluated this factor as a less important problem. The discriminability and applicability constraints as well as the limited energy of the respondents to follow the different information about sport nutrition were evaluated similarly. The item “I forget most information about sport nutrition right after I hear it” was not evaluated as a constraint of, at most, medium importance in the case of three-quarters of respondents. The descriptive characteristics of the responses to different items are summarized in Table 3. The relatively high values of skewness indicated a low probability of normality of distribution of responses; thus, we have tested the normality of individual responses by Kolmogorov–Smirnov and Shapiro–Wilk tests. Both tests have indicated the non-normality of distributions.

The differences in sport nutrition information overload by gender (Table 4) and level of qualification (Table 5) were analyzed. The level of acceptance of different items were practically the same in the case of men and women. Remembering sport nutrition information and feeling overloaded by the amount of sport nutrition information were perceived as more intensive barriers in the case of men than in the case of women.

In the case of qualification, we were not able to prove significant differences to at least a 10% level of significance. Sport nutrition information overload can be considered as a general problem among elite athletes, and it cannot be proven that this could be an important barrier only in athletes with lower education. Significance levels are marked with * at *p* < 0.05 and ** at *p* < 0.01.

The non-normal distribution of responses raises the issue of the multivariable normality of the dataset. This is a critical question because if the hypothesis of non-normal distribution is not supported, the further analyses can lead to standard error biases. We tested the multivariable normality hypotheses with five different tests, but none of them have supported the null hypothesis (Table 6). Under these conditions, we had to apply specific methods to obtain efficient test results; this is why we have applied Bollen–Stine analysis, specifically developed to address the problem of multivariable non-normality.

### 3.3. Validity Indicators

We determined the Cronbach α value, as a standard indicator of item validity. The indicators of reliability gave acceptable results—the sport nutrition information overload scale has high reliability and applicability. The scale reliability indices are summarized in Table 7.

The reliability of the scales was analyzed if the items were dropped out, one by one. The difference between these values was not more than 0.5%. The results of the simulation show that in all cases, the α is higher than 0.71, which is also considered as acceptable. The drop-out test of individual items highlighted that practically all items have similar importance on the scale. In Table 8, the standardized values are presented.

### 3.4. Unidimensionality Test by Confirmatory Factor Analysis

There was a relatively strong correlation between different items of the scale, but this coefficient was not higher than 0.57 in any pair of items. Figure 3 indicates that the highest correlation between items is 0.57 (absorption and discriminability constraints) and the lowest correlation is between memory and general overburden (0.23). The average interitem correlation is 0.37.

#### 3.4.1. Confirmatory Factor Analyses

There is a wide range of consensus on the applicability of confirmatory factor analysis on model fitting [23]. According to our basic hypothesis, the sport nutrition information overload scale is one-dimensional, thus we have tested it as such in this model. We have taken into consideration the non-normality of the data, which is why we have applied the Bollen–Stine bootstrap method [24] with *n* = 500 replications. The one latent variable test demonstrated acceptable results (Table 9). The different fit indices showed a relatively good fitting of the model, especially if we take into consideration the modest size of the sample. The RMSEA value was clustered to a 0.08 cutoff value, and the TLI value was near 0.9. Other additional measures of fitting have shown an acceptable fitting of a one-dimensional model, but have left room for specification errors. Thus, it was reasonable to continue the research by the determination of further, more sophisticated models.

#### 3.4.2. Bifactorial Confirmatory Factor Analysis

Based on confirmatory factor analysis, we were able to determine a model that is capable of describing a considerable part of the variance of the set under investigation. However, it is known that even when a general factor affects each item (in our case, information overload), there is a group of factors that accounts for one part of the residual variance [25].

Results of exploratory factor analysis have shown the presence of one common factor. Based on residuals, three factors could be determined. In the F1 factor, the only significant value has the memory constraint. General overload was significant in the F2 and F3 factors. This highlights that the general overload item is hardly suitable for the identification of information overload compared to other items. Evaluation of the confirmatory factor analysis results shows the low correlation between the F1 factor and another factor. The results, obtained by explanatory factor analysis to determine the fitting of the model, indicate a high level of reliability of the construction, but the factor structure is different. The general factor loadings (g) and factor loadings of residuals (F1, F2, F3) are summarized in Table 10 and Table 11.

Table 11 summarizes the most important stochastic relationships between the general factor and residual factors.

### 3.5. Exploratory Factor Loading Analysis

An alternative to McDonald’s ωh is the ωt. We have decomposed the variance of the test score into three components: (1) variance, explained by the general factor (g); (2) group factor variance, which is common to a given set of variables (items); and (3) the error terms. Theoretically, the specific variance and random error could have been distinguished, but this necessitates replicated measures, and in our case, this was not possible. The results of this test are summarized in Table 12 and Table 13. The exploratory bifactor analysis of variables yielded a different factor structure. In this case, the memory, applicability and energy constraints were the same factors. The time constraint was the only significant component of the F2 factor. Absorption and discriminability constraints were the components of the F3 factor. Interestingly, the attention and general overburden constraints could not be attached to any other items.

The results of the evaluation of exploratory factor analysis highlighted the stability of the model. This is supported by high values of ω in general for total scores and subscales (Table 13).

### 3.6. Validation of the Results by Hierarchical Cluster Analysis

Results were cross-validated by hierarchical cluster analysis. The item-clustering (Iclust) nesting clustering algorithm was applied to form hierarchical clusters in a stepwise manner until the internal consistency of aggregated clusters failed to increase. The first phase in the analysis of the scale’s structure was to estimate coefficients α and β for the overall scale. Both indicators were rather high, suggesting that a single scale is reliable. In the next step, we could separate two subscales, but in the C6 cluster of items, consisting of distinguishability, applicability, and time, the coefficients were relatively low. The other subcluster of scales had better indicators, but in this case, the general overload and the memory constraints can be evaluated as rather weak. Results of the hierarchical cluster analysis are summarized in Figure 4. The data on items include the average split-half reliability (α) and the global split-half reliability (β).

It is a commonly discussed problem if the number of questions on a scale or questionnaire could be reduced. Our results have proven that all of the items in the scale are relevant, as the weights of the items were approximately the same. Applying more sophisticated methods for the analysis of residuals, different patterns of similarity could be uncovered. These are summarized in Table 14. In this table, the same colors of cells indicate that the given items have been classified into the same group. The blank cells indicate that the given item did not have significant loading in a factor. Analyzing these patterns, three conclusions can be drawn: (1) Based on traditional indicators of reliability, all items were relevant. (2) The two bifactorial analyses and the rooted dendritic analysis showed different patterns, but the attention and time constraints, the applicability and energy constraints, as well as the absorption and discriminability constraints, have been grouped into the same factor/cluster two times out of three. For the classification of an item, measuring general information overload was rather hard because McDonald’s ωh was not able to assign this item to one factor. At the same time, the cluster analysis showed a considerable similarity between general information overload and the memory constraint. (3) The multivariable analysis was able to rather sharply separate the different patterns from each other.

## 4. Discussion

The information overload scale was adapted from the context of cancer patients and then it was validated to measure the perception of the information environment related to healthy eating [10]. Our study aimed to evaluate the applicability of the diet information overload scale and validate it among athletes. The original cancer-related information overload by Jensen et al. (2011) consists of eight items. Costa et al. (2014) shortened the original scale to a five-item scale [9,17]. Ramondt and Ramírez (2019) validated the original eight-item scale, but they recommended using the shortened version of the scale [10]. Our results highlight that all eight items of the scale are relevant based on traditional indicators of reliability; the original eight-item scale is suitable and reliable for the measurement of sport nutrition information overload.

By analyzing the results of the survey, it can be stated that sport nutrition information overload is a general problem among athletes. Athletes face a high quantity of sport nutrition information and feel like they have a relatively limited time to process it. Feeling overloaded by the amount of information about sport nutrition and remembering it have been perceived as stronger barriers in the case of men than in the case of women. A possible explanation for this can be the relatively lower level of interest of men in sport nutrition-related information. The lower interest in sport nutrition information could be associated with lower knowledge and misconceptions about sports nutrition [26].

On the contrary to the research results of Ramondt and Ramírez, our results highlight that sport nutrition information overload does not depend on the level of qualification. Athletes with lower and higher education both face this problem. This fact highlights the importance of understanding the intensity of sport nutrition information overload and the information flow of different social media platforms.

The “classic” indicators of reliability have given acceptable results. Taber (2018) analyzed the use and evaluation of Cronbach’s α when reporting different scales on the basis of 69 different papers [27]. He concluded that a Cronbach α of 0.81 has been evaluated as “high”, “good”, “adequate” or “reasonable” based on the opinion of the authors. If we take the results of the simulation, we can conclude that as the α is higher than 0.71, the high reliability and applicability of the scale are supported. The different fit indices have shown the relatively good fit of the model; the RMSEA value was clustered to the generally applied 0.08 cutoff value, and the TLI value was near 0.9, which is generally considered as an indicator of acceptable fit [28].

The socioeconomic structure of the sample is distorted because the higher qualified, urban part of the Hungarian athletes is over-represented, but this fact cannot be considered as a disqualifying limitation because our goal has been to test a questionnaire based on the absorption of sport nutrition-related pieces of information. Arguably, a more qualified part of the population is more suitable to supply adequate, coherent responses than such athletes to whom the decoding of simple, everyday communication would be a challenge.

The novelty of the current article is twofold: (1) we validate a tool for measurement of sport nutrition information overload in the case of athletes and examine athletes’ perceptions of the sport nutrition information environment; (2) we apply a combination of different sophisticated, cutting-edge validation methods, going beyond the traditional item reliability indicators (e.g., Cronbach α).

The limitation of the current validation study is that some participants may have been reluctant to report that they are not capable of keeping abreast with the flow of nutrition-related pieces of information. Theoretically, this could be a characteristic phenomenon in cases of such respondents whose profession is related to health, food, or nutrition.

## 5. Conclusions

Sport nutrition information overload is rather high among elite athletes. The sport nutrition information environment could influence athletes’ nutrition knowledge, health behavior, and could lead to misbeliefs regarding sport nutrition. Understanding of the nutrition information overload phenomenon and measuring athletes’ perceptions of the information environment related to sport nutrition are necessary preconditions of any sport nutrition-related intervention; thus, sophisticated, reliable tools need to be applied. Our results have proven that the Hungarian version of the sport nutrition information overload scale is suitable and reliable for practical application to study sport nutrition information overload among athletes.

## Figures and Tables

**Figure 1 nutrients-13-02781-f001:**
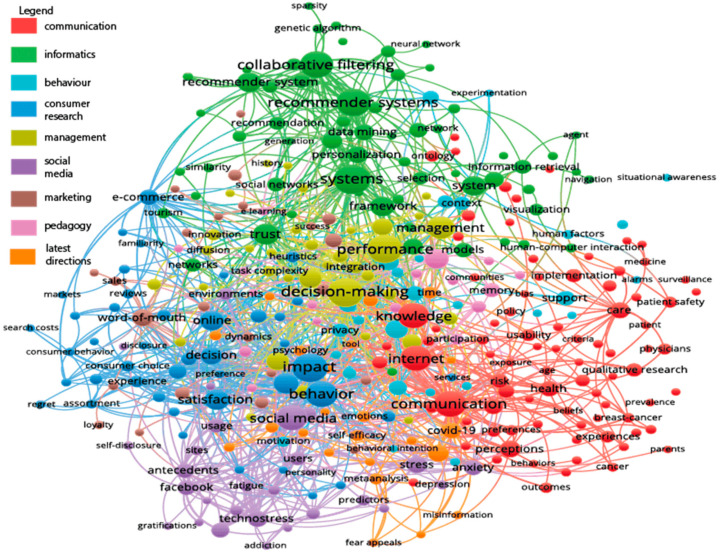
Main directions of information overload research.

**Figure 2 nutrients-13-02781-f002:**
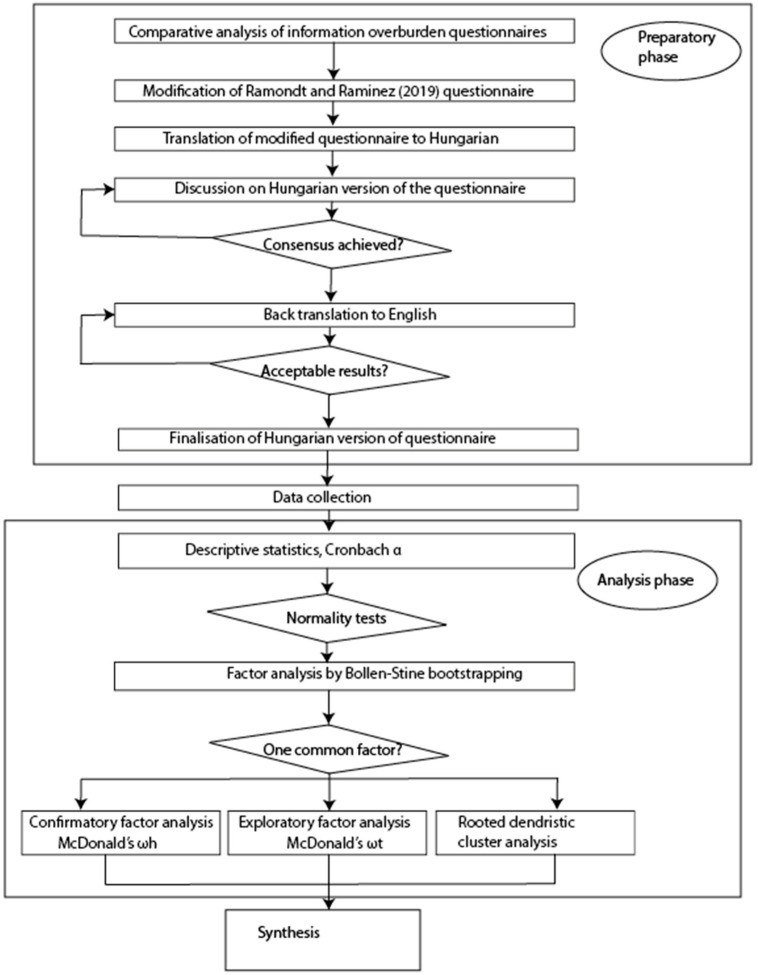
Workflow of data analysis. The rectangles show different steps, built on each other, and the rhomboids—in line with the general practice of block-diagram building—symbolize the decision points, in some cases, the hooks and cycles.

**Figure 3 nutrients-13-02781-f003:**
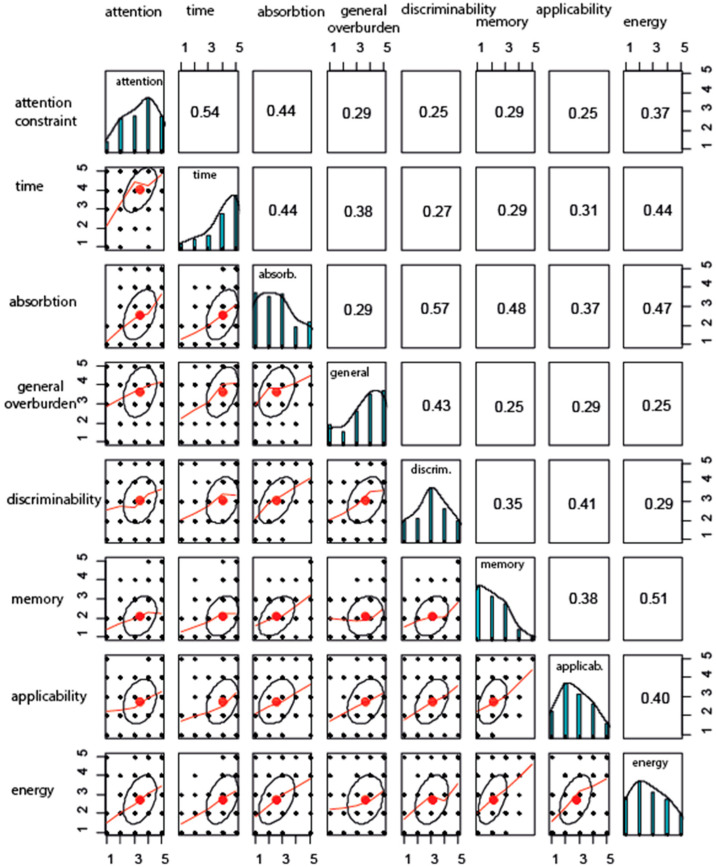
The correlational structure of items.

**Figure 4 nutrients-13-02781-f004:**
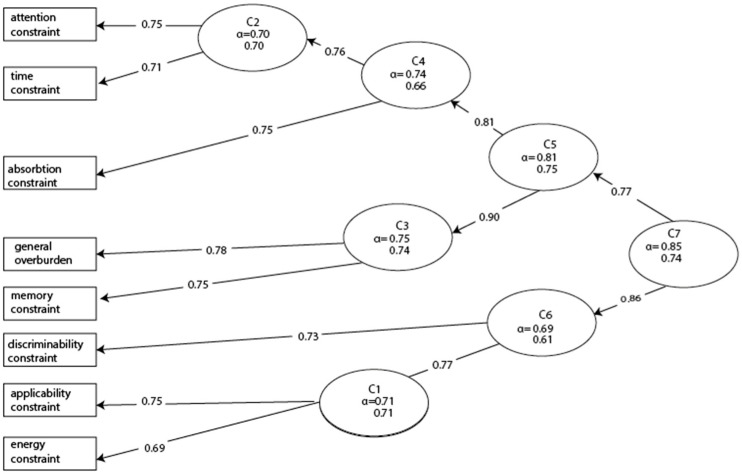
Rooted dendritic structure of statements.

**Table 1 nutrients-13-02781-t001:** Information overload scale for different focus groups.

Original Cancer Information Overload [9]	Short Form of Cancer Information Overload [17]	Public Diet Information Overload [10]	Sport Nutrition Information Overload (Kiss et al., 2021), (in Square Brackets We Have Indicated the Shortened Name of the Item, Applied in the Current Article)
There are so many different recommendations about preventing cancer, it’s hard to know which ones to follow.	There are so many different recommendations about preventing cancer, it’s hard to know which ones to follow.	There are so many recommendations about eating a healthy diet, it’s hard to know which ones to follow.	There are so many recommendations about sport nutrition, it’s hard to know which ones to follow [attention constraint].
There is not enough time to do all of the things recommended to prevent cancer.	-	There is not enough time to do all the things recommended to eat a healthy diet.	There is not enough time to do all the things recommended eating for sports [time constraint].
It has gotten to the point where I don’t even care to hear new information about cancer.	It has gotten to the point where I don’t even care to hear new information about cancer.	It has gotten to the point where I don’t even care to hear new information about eating a healthy diet.	It has gotten to the point where I don’t even care to hear new information about sport nutrition [perception constraint].
No one could actually do all of the cancer recommendations that are given.	-	No one could actually do all of the healthy diet recommendations that are given.	No one could actually do all of the sport nutrition recommendations that are given [general overload].
Information about cancer all starts to sound the same after a while.	Information about cancer all starts to sound the same after a while.	Information about eating a healthy diet all start to sound the same after a while.	Information about sport nutrition all start to sound the same after a while [discriminability constraint].
I forget most cancer information right after I hear it.	I forget most cancer information right after I hear it.	I forget most information about eating a healthy diet right after I hear it.	I forget most information about sport nutrition right after I hear it [memory constraint].
Most things I hear or read about cancer seem pretty far-fetched.	-	Most things I hear or read about eating a healthy diet seem pretty far-fetched.	Most things I hear or read about sport nutrition seem pretty far-fetched [applicability constraint].
I feel overloaded by the amount of cancer information I am supposed to know.	I feel overloaded by the amount of cancer information I am supposed to know.	I feel overloaded by the amount of information about eating a healthy diet I am supposed to know.	I feel overloaded by the amount of information about sport nutrition I am supposed to know [energy constraint].

**Table 2 nutrients-13-02781-t002:** Basic sociodemographic characteristics of the sample.

Sociodemographic Characteristics	Distribution
Gender	
Male	51.9%
Female	49.1%
Highest accomplished qualification	
Elementary school	12.3%
Secondary school	32.5%
At least BSc/BA	66.2%
Place of living	
Capital city	38.1%
City	31.1%
Small town	25.8%
Village	5%
Average age of respondents (years)	29.4
Sport activity	
Average years, spend sporting (years)	16.9
Average number of weekly training	9.3
Average number of sport competitions in a year	17.4
Types of sport	
Swimming	21%
Rowing	16.7%
Volleyball	14%

**Table 3 nutrients-13-02781-t003:** Descriptive statistics of responses to different items.

Items of the Scale	Mean	Standard Deviation	Skewness	Kurtosis
There are so many recommendations about sport nutrition, it’s hard to know which ones to follow.	3.43	1.24	−0.32	−0.91
There is not enough time to do all the things recommended eating for sports.	4.07	1.19	−1.06	0.20
It has gotten to the point where I don’t even care to hear new information about sport nutrition.	2.54	1.27	0.49	−0.84
No one could actually do all of the sport nutrition recommendations that are given.	3.62	1.33	−0.73	−0.52
Information about sport nutrition all start to sound the same after a while.	3.07	1.27	−0.12	−0.74
I forget most information about sport nutrition right after I hear it.	2.07	1.04	0.67	−0.31
Most things I hear or read about sport nutrition seem pretty far-fetched.	2.69	1.12	0.24	−0.87
I feel overloaded by the amount of information about sport nutrition I am supposed to know.	2.73	1.22	0.22	−1.05

**Table 4 nutrients-13-02781-t004:** Average and standard deviation of responses by gender. Significance levels are marked with * at *p* < 0.05 and ** at *p* < 0.01.

Items of the Scale	Men	Women	Significant Difference
There are so many recommendations about sport nutrition, it’s hard to know which ones to follow.	3.64 ± 1.15	3.21 ± 1.23	
There is not enough time to do all the things recommended eating for sports.	4.15 ± 1.14	3.88 ± 1.2	
It has gotten to the point where I don’t even care to hear new information about sport nutrition.	2.8 ± 1.34	2.35 ± 1.27	
No one could actually do all of the sport nutrition recommendations that are given.	3.64 ± 1.39	3.63 ± 1.25	
Information about sport nutrition all start to sound the same after a while.	3.09 ± 1.26	3.04 ± 1.18	
I forget most information about sport nutrition right after I hear it.	2.33 ± 1.1	1.86 ± 0.95	**
Most things I hear or read about sport nutrition seem pretty far-fetched.	2.74 ± 1.17	2.72 ± 1.19	
I feel overloaded by the amount of information about sport nutrition I am supposed to know.	2.92 ± 1.27	2.56 ± 1.27	*

**Table 5 nutrients-13-02781-t005:** Average and standard deviation of responses by qualification.

**Items of the Scale**	**Elementary School**	**Secondary School**	**At Least BSc**
There are so many recommendations about sport nutrition, it’s hard to know which ones to follow.	3.48 ± 1.23	3.26 ± 1.21	3.5 ± 1.09
There is not enough time to do all the things recommended eating for sports.	4.04 ± 1.17	3.88 ± 1.25	4.21 ± 0.97
It has gotten to the point where I don’t even care to hear new information about sport nutrition.	2.66 ± 1.35	2.38 ± 1.29	2.57 ± 1.22
No one could actually do all of the sport nutrition recommendations that are given.	3.63 ± 1.26	3.67 ± 1.39	3.5 ± 1.45
Information about sport nutrition all start to sound the same after a while.	3.06 ± 1.25	3.07 ± 1.16	3.07 ± 1.27
I forget most information about sport nutrition right after I hear it.	1.99 ± 1.09	2.24 ± 0.98	2.21 ± 0.97
Most things I hear or read about sport nutrition seem pretty far-fetched.	2.78 ± 1.2	2.55 ± 1.06	3.00 ± 1.36
I feel overloaded by the amount of information about sport nutrition I am supposed to know.	2.85 ± 1.4	2.57 ± 1.09	2.5 ± 1.02

**Table 6 nutrients-13-02781-t006:** Results of multivariable normality test.

Indicator	Value of Test Statistics	Significance
Mardia skewness	271.127	0.000
Mardia Kurtosis	1.761	0.073
Henze–Zirkler test	HZ = 1.068	0.000
Royston test	H = 351.337	0.000
E-statistics	1.897	0.000

**Table 7 nutrients-13-02781-t007:** Traditional scale reliability indices.

Indicators	Value
α based upon the covariances	0.822
standardized α based upon the correlations	0.819
Guttman’s Lambda 6 reliability	0.816

**Table 8 nutrients-13-02781-t008:** Reliability if items are dropped, one by one.

Items of the Scale	Cronbach α	Guttman’s λ6
There are so many recommendations about sport nutrition, it’s hard to know which ones to follow.	0.789	0.795
There is not enough time to do all the things recommended eating for sports.	0.795	0.784
It has gotten to the point where I do not even care to hear new information about sport nutrition.	0.769	0.777
No one could actually do all of the sport nutrition recommendations that are given.	0.899	0.897
Information about sport nutrition all start to sound the same after a while.	0.801	0.798
I forget most information about sport nutrition right after I hear it.	0.805	0.799
Most things I hear or read about sport nutrition seem pretty far-fetched.	0.799	0.798
I feel overloaded by the amount of information about sport nutrition I am supposed to know.	0.786	0.787

**Table 9 nutrients-13-02781-t009:** Fit indices and variable coefficients of one latent variable confirmatory factor analysis model.

Fit Indices	Value
Comparative Fit index (CFI)	0.901
Tucker-Lewis index (TLI)	0.853
Root Mean Square Error of Approximation (RMSEA)	0.087
Standardized Rood Mean Square Residual (SRMR)	0.059
gammahat	0.939
adj. gammaHat	0.897
baseline RMSEA	0.269
AIC smallN	3421
AIC smallN	3500
BIC priorN	3741
hqc	3351

**Table 10 nutrients-13-02781-t010:** Results of confirmatory factor loading analysis by Schmid–Leiman method.

Items of the Scale	g	F1	F2	F3	h2	p2
There are so many recommendations about sport nutrition, it’s hard to know which ones to follow.	0.55		0.39		0.41	0.63
There is not enough time to do all the things recommended eating for sports.	0.53		0.72		0.78	0.37
It has gotten to the point where I don’t even care to hear new information about sport nutrition.	0.64			0.27	0.53	0.76
No one could actually do all of the sport nutrition recommendations that are given.	0.37		0.29	0.26	0.31	0.44
Information about sport nutrition all start to sound the same after a while.	0.52			0.87	0.96	0.23
I forget most information about sport nutrition right after I hear it.	0.63	0.31			0.50	0.75
Most things I hear or read about sport nutrition seem pretty far-fetched.	0.57				0.32	0.84
I feel overloaded by the amount of information about sport nutrition I am supposed to know.	0.69	0.34			0.58	0.83

The h2 column shows the amount of variance of the item, explained by the retained factors; p2 is the ratio between g2/h2, which can be considered as a diagnostic tool.

**Table 11 nutrients-13-02781-t011:** Evaluation of confirmatory factor analysis by Schmid–Leiman method.

Indicators	g	F1	F2	F3
correlation of scores with factors	0.85	0.41	0.79	0.91
Multiple R2 of scores with factors	0.73	0.22	0.63	0.89
Minimum correlation of factor score estimates	0.42	−0.612	0.23	0.75
ω total for total scores and subscales	0.84	0.69	0.66	0.83
ω general for total scores and subscales	0.59	0.61	0.32	0.42
ω group for total scores and subscales	0.12	0.15	0.37	0.41

**Table 12 nutrients-13-02781-t012:** Results of exploratory factor loading analysis by Schmid–Leiman method.

Items of the Scale	g	F1	F2	F3	h2	p2
There are so many recommendations about sport nutrition, it’s hard to know which ones to follow.	0.56				0.35	0.94
There is not enough time to do all the things recommended eating for sports.	0.58		0.21		0.86	0.21
It has gotten to the point where I don’t even care to hear new information about sport nutrition.	0.76			0.41	0.61	0.87
No one could actually do all of the sport nutrition recommendations that are given.	0.47				0.26	0.95
Information about sport nutrition all start to sound the same after a while.	0.56			0.34	0.43	0.77
I forget most information about sport nutrition right after I hear it.	0.57	0.31			0.39	0.77
Most things I hear or read about sport nutrition seem pretty far-fetched.	0.51	0.25			0.37	0.81
I feel overloaded by the amount of information about sport nutrition I am supposed to know.	0.66	0.45			0.61	0.65

The h2 column shows the amount of variance of the item, explained by the retained factors; p2 is the ratio between g2/h2, which can be considered as a diagnostic tool.

**Table 13 nutrients-13-02781-t013:** Evaluation of exploratory factor analysis by Schmid–Leiman method.

Indicator	g	F1	F2	F3
correlation of scores with factors	0.86	0.59	1.52	0.48
Multiple R2 of scores with factors	0.79	0.34	2.25	0.27
Minimum correlation of factor score estimates	0.54	−0.32	3.47	−0.49
ω total for total scores and subscales	0.91	0.74	0.91	0.72
ω general for total scores and subscales	0.76	0.53	0.48	0.55
ω group for total scores and subscales	0.16	0.21	0.45	0.31

**Table 14 nutrients-13-02781-t014:** Patterns of connections between items, determined by different methods (same colors indicate similarity).

Items of the Scale	Cronbach α and Guttman’s λ_6_	McDonald’s ω_h_	McDonald’s ω_t_	Rooted Dendritic Structure Analysis
There are so many recommendations about sport nutrition, it’s hard to know which ones to follow.				
There is not enough time to do all the things recommended eating for sports.				
It has gotten to the point where I don’t even care to hear new information about sport nutrition.				
No one could actually do all of the sport nutrition recommendations that are given.				
Information about sport nutrition all start to sound the same after a while.				
I forget most information about sport nutrition right after I hear it.				
Most things I hear or read about sport nutrition seem pretty far-fetched.				
I feel overloaded by the amount of information about sport nutrition I am supposed to know.				

## Data Availability

The data presented in this study are available on request from the corresponding author. The data are not publicly available due to data protection requirements.

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
