# Peer review of "Measuring Athletes’ Perception of the Sport Nutrition Information Environment: The Adaptation and Validation of the Diet Information Overload Scale among Elite Athletes"

_nutrients, 2021, doi:10.3390/nu13082781_

Round 1

Reviewer 1 Report

  1. The abstract is not informative enough. The short description of diet information overload scale should be given.
  2. The studied group consisted of licensed athletes (74.2%) and recreational athletes (25.8%). The results should be analysed separately for licensed athletes and recreational athletes, and it should be considered the title of the manuscript, study hypotheses, discussion, and conclusions. Or recreational athletes should be removed from the study and analysis.
  3. The study is very well designed, and the statistical analysis is impressive and outstanding. However, in my opinion the manuscript is beyond the scope of the special issue entitled Sport Nutrition Knowledge of Athletes and Implications for Dietary Habits, Nutrient Status and Energy Availability, while no relations between nutritional knowledge and dietary habits, nutrient status or energy availability are given.

Reviewer 2 Report

This is an interesting and useful work, but the article is too long that rambling.

Please, work on the structure of the article. All info provided is mixed-up, particularly in the second part.

Introduction. The authors talked at length for topics, which are not relative to the research question. The authors need to narrow their focus in the specific topic of their study in the introduction and get to the point. The lines that refer to the information and the information overload seem to be a philosophical review and should be omitted. The introduction should preferably start somewhere from line 78 and still many lines thereafter can be omitted.

Objective: Please stick to the aim of the study. Lines 116-121 that refer to the novelty should be mentioned after the presentation of the authors’ work and not beforehand. This can possibly be in the discussion part, before limitations

In the 2nd  part (Materials and Method) authors should start with the participants and thereafter should describe the study design, the method used, including the questionnaire and end up with the statistical analysis.

Concerning the participants, the recruitment and inclusion and exclusion criteria should be described in this part. The number of the participants and their characteristics (lines 177-199) should be moved into the results. Was there an ethics approval and informed consent?  

Results include the analyses used for the validity steps. However, the answers of the participants on the scale are not presented. It would be interesting to present and analyze their answers according to the age, sex, years of training, different disciplines and socio-demographic characteristics.

The discussion focuses on the statistical analysis, the adaptation and validation of the scale and not on the results of the scale answered by the athletes. The conclusions do not derive from the study, but this part also includes a long rambling speech.

Round 2

Reviewer 1 Report

I appreciate and accept all the revisions

Reviewer 2 Report

I am pleased with the changes. The article can be published in its present form